# Adolescent mental health difficulties and educational attainment: findings from the UK household longitudinal study

Neil R Smith [ID],[1] Lydia Marshall,[1] Muslihah Albakri,[1] Melanie Smuk,[2] Ann Hagell,[3] Stephen Stansfeld [ID] [4]

¹Policy Research Centre, NatCen Social Research, London, UK
²London School of Hygiene & Tropical Medicine, London, UK
³Association for Young People's Health, London, UK
⁴Psychiatry, Barts and the London, Queen Marys School of Medicine and Dentistry, London, UK

**Correspondence to**
Dr Neil R Smith;
neil.smith@natcen.ac.uk

## ABSTRACT

**Objective** This study examines whether there is an independent association between mental difficulties in adolescence and educational attainment at age 16.

**Design** Longitudinal study.

**Setting** Nationally representative data from the UK Household Longitudinal Study (UKHLS) were linked to the National Pupil Database for England.

**Participants** Respondents (N=1100) to the UKHLS between 2009 and 2012 were linked to the National Pupil Database to investigate longitudinal associations between mental difficulties at ages 11–14 and educational attainment at age 16 (General Certificate of Secondary Education (GCSE)).

**Primary outcome measure** Not gaining five or more GCSE qualifications at age 16, including English and maths at grade A*–C.

**Results** An atypical total mental health difficulty score measured using the Strengths and Difficulties Questionnaire at ages 11–14 predicted low levels of educational attainment at age 16 (OR: 3.11 (95% CI: (2.11 to 4.57)). Controlling for prior attainment and family sociodemographic factors, happiness with school (/work) and parental health, school engagement and relationship with the child partially attenuated the association, which was significant in the fully adjusted model (2.05, 95% CI (1.15 to 3.68)). The association was maintained in the fully adjusted model for males only (OR: 2.77 (95% CI (1.24 to 6.16)) but not for females. Hyperactivity disorder strongly predicted lower attainment for males (OR: 2.17 (95% CI: (1.11 to 4.23)) and females (OR: 2.85 (95% CI (1.30 to 6.23)).

**Conclusion** Mental difficulties at ages 11–14 were independently linked to educational success at age 16, highlighting an important pathway through which health in adolescence can determine young people's life chances.

## INTRODUCTION

Growing evidence of the prevalence of poor child and adolescent mental health has led to this issue becoming a key policy priority in the UK. Mental ill-health in children and young people in England increases age with around 14.4% of 11–16 years experiencing a mental disorder compared with 5.5% in

### Strengths and limitations of this study

► This is a large, nationally representative longitudinal cohort study containing self-assessed measures of mental health among young people linked to a National Pupil Database of educational records.

► The study captures a diverse range of social, demographic, economic and behavioural factors affecting young people in their home and school environment, permitting statistical adjustment for multiple confounding relationships, which might explain the association between mental health and educational attainment.

► Consent to data linkage between the longitudinal study and the National Pupil Database was incomplete, though factors which predicted patterns of nonconsent were controlled for within our models.

► Missing data were accounted for using multiple imputation methods, which exploited the wide range of associations within the observed data to minimise errors within estimates of effect.

their preschool counterparts aged 2–4 years.[1] With 75% of adult mental health problems (excluding dementia) starting by the age of 18,[2] adolescence is a key period in the development of long-lasting mental health difficulties. The UK government's *Future in Mind* report[2] presented an important economic case for investment in early prevention of mental ill-health to mitigate against the costs of longer term support for health needs. However, this argument neglects the impact that early life mental health potentially has on other early life outcomes fundamental in determining life chances, such as educational attainment.[3] Educational outcomes are closely associated with later-life chances with well-established links to employment, income, housing and offending as well as physical health and on-going mental health disorders. If poor mental health diminishes the capacity for individuals to fulfil their

academic potential, mental health itself is likely to be a driver of educational inequality and consequent on-going social inequality.

On the other hand, the association between mental health and educational outcomes might not be direct, but rather incorporate the influence of confounding factors. A range of demographic and socioeconomic factors, such as gender, ethnicity, socioeconomic disadvantage and maternal education and parental health,[4–6] have known relationships with educational attainment and must be accounted for when assessing the impact of poor mental health. Similarly, the home environment and specifically parental interest in schooling have been associated with higher attainment,[7] as have positive environmental 'school effects',[8] whereas lower attainment has been associated with absence from school[1] or poor classroom behaviours.[9] What is less clear is the extent to which differential exposure to these factors also underpins disparities in mental health, and whether resulting differences in mental health might influence differences in attainment.

International research has demonstrated numerous associations between mental health and educational attainment.[10–12] The evidence base for England is less well established, which is of particular relevance during a time of policy interest in boosting mental health provision in schools.[13] There is some evidence of longitudinal associations between psychological distress in early adolescence and achievement at General Certificate of Secondary Education (GCSE) in England.[14 15] Similarly, poor mental health between ages 13 and 15 has been shown to be associated with low GCSE attainment and later unemployment,[9] demonstrating how the effects of poor early life mental health can extend into adulthood.[16] Though many of these findings support the association between mental health and educational outcomes, they are often of low generalisability being based on regional data or nonprobability samples[14] or unable to account for a range of potentially explanatory factors.[15] There appears to be a strengthening of the relationship between adolescent mental health and educational outcomes in recent generations,[17] so there is a pressing need for an up-to-date examination of nationally representative data for England.

Therefore, this study uses a novel and contemporary data linkage between the nationally representative UK Household Longitudinal Study (UKHLS) linked to objectively measured official education records, to test associations between poor mental health and poor educational attainment. The study is significant in estimating the extent to which mental health in early adolescence has an independent association with attainment at age 16 in England in males and females. Robust evidence of a causal relationship between poor mental health and lower academic attainment could be crucial in inspiring investment in researching 'what works' in supporting children and adolescents' mental health. Although schools already appreciate the importance of supporting pupils' health and well-being,[18] a proven link to academic outcomes could also encourage education and public health policy-makers to invest more in mental health.

## METHODS

### UK Household Longitudinal Study

The UKHLS is a nationally representative household panel survey,[19] which began in 2009, aiming to understand social and economic change in Britain at the household and individual levels. Each wave of the survey collects information on approximately 100 000 individuals from 40 000 households, with adult household residents (aged 16 and over) responding using computer-assisted interview and self-completion questionnaire. Young people aged between 10 and 15 were offered a self-completion questionnaire. Further details on the sampling design and data collection are available.[20] National educational records from the National Pupil Database (NPD)[21] for school-age children between ages 3 and 18 were linked to the UKHLS if parents and their children were living in England and consented to linkage at wave 1. Linkage consent rates did not differ systematically by parental class, or parental education though they were lower within ethnic minority groups, which is consistent with other cohort studies.[22]

This analysis used a nationally representative sample of 11–14-year olds present at wave 1 (2009–2011) and wave 3 (2011–2013) linked to the NPD. Wave 2 (2010–2012) was excluded as it did not ask for information about mental health. Where respondents were present at both waves, data from wave 3 were selected as the respondent was further into adolescence. Figure 1 tracks the study population down to the final analytical sample.

The final sample consisted of all consenting youth panel respondents aged 11–14 years with data on mental health in wave 1 or wave 3 of UKHLS as well as NPD data on GCSE scores at ages 15 or 16 years (N=1110). The analytic sample covers England only due to the limited geographical coverage of the NPD.

### Educational attainment

The primary outcome was a binary variable indicating low educational attainment, defined as whether the young person did not achieve five or more grades A*–C for the GCSE, including English and maths. This was the benchmark measure of educational attainment at key stage 4 (KS4) at secondary schools in England during the study period.[23]

### Mental difficulties

Young people completed the Strengths and Difficulties questionnaire (SDQ) validated for ages 4–15 years.[24] The SDQ asks questions about four domains of negative behaviours that have varying strengths of association with educational attainment, namely, conduct problems,[11] hyperactivity,[25] emotional symptoms,[14] peer problems.[26] Scores from the four subscales were summed to construct

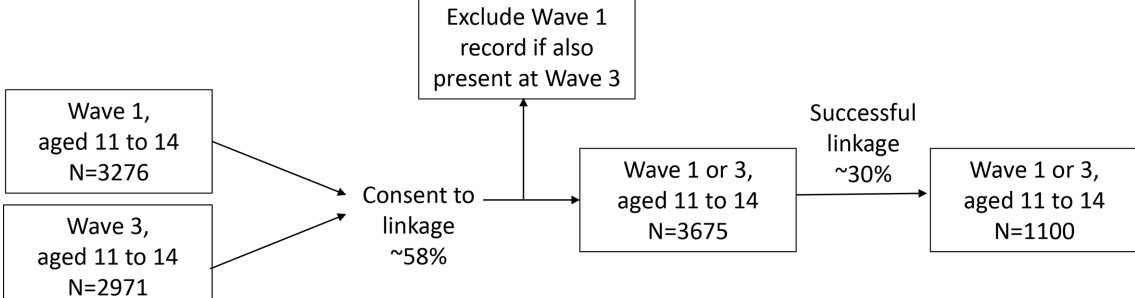

**Figure 1** Flow chart describing the breakdown of the combined wave 1 and wave 3 study population of the UKHLS into the analytic sample. UKHLS,UK Household Longitudinal Study.

a total difficulty score, where a higher score refers to a greater level of mental difficulties. Binary measures of mental difficulties were derived based on developer guidance.[24] An 'atypical' level of total difficulties was derived from the top 10% of the population scores (>=18 out of 40) and individual SDQ domains used validated 'atypical' cut points that have also been used in a recent prevalence survey in England.[27]

### Explanatory variables

We focused on risk factors where the literature has established potentially causal associations with educational attainment and mental health, respectively. All analyses were controlled for gender, age, ethnic group as well as the three-tiered classification household's highest parental occupational class, household deprivation and mother's highest educational qualifications.[28–30] Parents' highest current or previous occupational class was based on the National Statistics Socioeconomic Classification (NS-SEC), which was collapsed into a three-tier hierarchical scale (professional/managerial; intermediate; manual/routine)[31] with an additional category for overseas or no qualifications. The mother's highest qualification was summarised on a three-tier hierarchical scale (degree or higher, A-level or equivalent, GCSE or equivalent with a separate category for none or other). Household poverty was derived based on income poverty, material poverty, subjective poverty and the receipt of benefits and was categorised into 'not at all deprived', 'somewhat deprived' or 'highly deprived'.[32] Additionally, family type was grouped into two parent households, lone parent household or other family types.[28]

Parental relationships were assessed using a binary measures of young people's self-reports on how interested their parent(s) were at how they did at school, attendance at parents' evenings, frequency of quarrelling with either parent(s) and how often they feel supported by their family.[7] Parental physical and mental health was assessed using the SF-12 Physical and Mental Component Summary, respectively,[33] with a score from either parent in the lowest quintile representing poor physical health and a mental health score of ≥45.6 representing poor mental health.

Young people reported levels of happiness specifically with school work as well as with school generally on a

7-point scale with a score of 5 or greater indicating happiness.[34] Prior attainment was measured based on whether young people achieved the expected level 4 reading, writing and mathematics at KS2 (ages 10–11 years).

All non-educational attainment measures were taken at the time adolescent mental health was assessed.

### Statistical analysis

Complete data were available for age, sex, ethnicity and family composition. Missing data were most common for household poverty (13%) so data were imputed under the missing at random assumption as poverty was associated with poorer explanatory outcomes, specifically lower level of occupational class, maternal education, family composition and prior attainment. Given the overall low level of missingness, 20 imputed datasets were created. All explanatory variables and measures of mental difficulties shown in table 1 were used in the imputation and missing data for explanatory variables (ranging between 1% and 13%) and mental difficulties (0.2%) was imputed. Data on GCSE grades were not imputed due to a high proportion of missing data (70%) due to a lack of linkage consent, and for ethical reasons given these individuals had not consented to their data being used for research.

The prevalence of low attainment and mental difficulties are described separately according to a range of selected socioeconomic, demographic and parent-related factors. Data were weighted using the cross-sectional self-completion weights in the UKHLS youth panel in wave 1 and wave 3.

Logistic regression was used to estimate separately the OR of not achieving 5 A*–C GCSE grades including English and mathematics and of being classed as having mental difficulties. Stepwise regression models adjusted the ORs of having total mental difficulties and difficulties within each domain to examine the relative impact of prior attainment, sociodemographic factors, parent–child relationships, young person's happiness with school and parental health on educational attainment. Models were stratified to explore gender differences in total and domain specific mental difficulties. All analyses were performed in Stata V.16.1 (StataCorp, College Station, Texas).

**Table 1** Prevalence % of low educational attainment at key stage 4 by sociodemographic and parental characteristics

| | % (N) | Low attainment % | OR | 95% CI |
|---|---|---|---|---|
| **Sex** | | | | |
| Male | 51.6 (550) | 42.0 | 1 | Ref |
| Female | 48.4 (560) | 31.5*** | 0.64*** | (0.49 to 0.83) |
| **Age (years)** | | | | |
| 11 | 1.1 (14) | 65.5* | 3.42* | (1.05 to 11.15) |
| 12 | 9.7 (111) | 38.4 | 1.12 | (0.72 to 1.76) |
| 13 | 38.9 (432) | 37.3 | 1.07 | (0.81,1.42) |
| 14 | 50.4 (553) | 35.7 | 1.00 | Ref |
| **Ethnic group** | | | | |
| White British | 86.1 (839) | 36.9 | 1 | Ref |
| Other ethnic group | 13.9 (271) | 37.0 | 1.00 | (0.72 to 1.40) |
| **Parental highest social class (NS-SEC)** | | | | |
| Management and professional | 41.8 (439) | 23.4 | 1.00 | Ref |
| Intermediate | 22.7 (253) | 34.2** | 1.70** | (1.19 to 2.44) |
| Routine and manual | 31.0 (345) | 53.6** | 3.79*** | (2.74 to 5.25) |
| Unemployed | 4.4 (53) | 61.3** | 5.18*** | (2.60 to 10.35) |
| **Mother's highest qualification** | | | | |
| Degree or higher | 33.2 (351) | 24.0 | 1 | Ref |
| A-level or equivalent | 17.5 (185) | 21.8 | 0.88 | (0.57 to 1.38) |
| GCSE or equivalent | 29.5 (309) | 41.3*** | 2.23*** | (1.57 to 3.19) |
| None/other | 19.8 (239) | 65.4*** | 6.00*** | (4.06 to 8.86) |
| **Household poverty score** | | | | |
| Not at all deprived | 20.9 (179) | 16.2 | 1 | Ref |
| Somewhat deprived | 54.0 (493) | 35.8*** | 2.89*** | (1.84 to 4.56) |
| Highly deprived | 25.1 (266) | 56.5*** | 6.74*** | (4.08 to 11.13) |
| **Family composition** | | | | |
| Two parent | 69.7 (759) | 32.9 | 1 | Ref |
| Single parent | 27.8 (321) | 47.6*** | 1.86*** | (1.39 to 2.47) |
| Other | 2.5 (30) | Suppressed | – | – |
| **Happy with school work** | | | | |
| Happy | 74.7 (840) | 29.6 | 1 | Ref |
| Not happy | 25.3 (263) | 58.6*** | 3.38*** | (2.49 to 4.57) |
| **Happy with school** | | | | |
| Happy | 78.6 (876) | 32.0 | 1 | Ref |
| Not happy | 21.4 (220) | 54.7*** | 2.57*** | (1.86 to 3.53) |
| **Parental interest in school** | | | | |
| Always or nearly always | 79.0 (871) | 34.4 | 1 | Ref |
| Sometimes or rarely | 21.0 (220) | 46.4** | 1.66** | (1.20 to 2.28) |
| **Regularly attends parents' evenings** | | | | |
| Always or nearly always | 81.1 (896) | 29.6 | 1 | Ref |
| Sometimes or rarely | 18.9 (199) | 68.0*** | 5.05*** | (3.56 to 7.16) |
| **Feels supported by family** | | | | |
| Always or mostly | 76.3 (837) | 34.7 | 1 | Ref |
| Not supported | 23.7 (269) | 44.1* | 1.49* | (1.10 to 2.02) |
| **Regularly quarrels with either parent** | | | | |

**Table 1** Continued

| | % (N) | Low attainment % | OR | 95% CI |
|---|---|---|---|---|
| Less than once a week | 60.0 (662) | 33.1 | 1 | Ref |
| More than once a week | 40.0 (423) | 42.6** | 1.50** | (1.14 to 1.97) |
| Parental mental health | | | | |
| Not poor | 56.8 (539) | 30.0 | 1 | Ref |
| Poor | 43.2 (423) | 46.0*** | 1.98*** | (1.50 to 2.62) |
| Parental physical health | | | | |
| Not poor | 58.6 (564) | 32.9 | 1 | Ref |
| Poor | 41.4 (402) | 42.6** | 1.52** | (1.15 to 2.00) |
| Attainment at key stage 2 Maths | | | | |
| Achieved level 4 | 71.5 (860) | 26.6 | 1 | Ref |
| Did not achieve level 4 | 17.4 (169) | 85.9*** | 16.92*** | (10.65 to 26.87) |
| Attainment at key stage 2 Writing | | | | |
| Achieved level 4 | 82.6 (270) | 22.2 | 1 | Ref |
| Did not achieve level 4 | 28.4 (759) | 73.9*** | 9.96*** | (7.14 to 13.90) |
| Attainment at key stage 2 Reading | | | | |
| Achieved level 4 | 92.3 (947) | 32.4 | 1 | Ref |
| Did not achieve level 4 | 7.7 (74) | 91.5*** | 22.65*** | (9.85 to 52.09) |

Ref=reference group; unweighted N; imputed and weighted percentages shown; low educational attainment defined as <5 GCSEs at A*–C including English and maths; some values are suppressed due to small base sizes and risk of disclosure; *** p<0.001, **p<0.01, *p<0.05.
GCSE, General Certificate of Secondary Education.

## RESULTS

The analytic sample was evenly split by gender and the overwhelming majority were aged 13 or 14 years old. Respondents tended to be from relatively socioeconomically advantaged backgrounds. A third of mothers were degree educated and 41.8% of households belonged to the highest social class. Over three-quarters of the sample reported high parental engagement with school and happiness with school work. Prior attainment levels were positive for reading (93.3%), writing (82.6%) and maths (71.5%).

The proportion of young people not achieving the KS4 benchmark of 5 GCSEs A*–C including English and maths varied by selected characteristics (table 2). Low prior attainment at KS2 was most strongly associated with not reaching the educational benchmark at KS4. Low attainment at KS4 was also associated with lower social class, lower maternal education, higher household poverty scores and poorer parent–child relationships as well as poor parental mental and physical health. Reported unhappiness with school and school work and lower parental involvement in schooling were also significantly associated with low attainment.

There was a similar pattern to the prevalence of mental difficulties (table 2). Poorer household socioeconomic circumstances, parental engagement with school and health, parent–child relationships and the young person's happiness with school and school work were all significantly associated with increased odds of being classified with mental difficulties. However, there was no significant difference in the prevalence of mental health difficulties by sex, and the association between prior attainment and current mental difficulties was relatively weak and significant only for writing at KS2.

Young people classified with mental health difficulties were over three times more likely to not reach the KS4 GCSE benchmark (OR 3.11, 95% CI (2.11 to 4.57)) in the unadjusted model (table 3). Incrementally controlling for prior attainment and household socioeconomic factors did not attenuate this risk. Controlling for a young person's happiness with school and school work (model 5) and parental relationships and support (model 6) partially diminished this risk. However, the fully adjusted model demonstrated that young people with poor mental health were over two times as likely (OR 2.05, 95% CI (1.15 to 3.68)) to not reach the educational benchmark than their counterparts with subclinical difficulties. Within individual subdomains, the fully adjusted model could not account for the higher odds of not reaching the educational benchmark for those with hyperactivity disorder (OR 2.38, 95% CI (1.48 to 3.82)), implying that hyperactivity disorder largely drives the association between mental difficulty scores and lower attainment. For emotional and peer disorders, these risks were no longer significant once adjusted for prior attainment and sociodemographic factors, and conduct disorder no longer predicted lower attainment following adjustment for happiness with school and school work.

Table 4 describes the sex-specific association between mental health difficulties and attainment to explore the

**Table 2** Prevalence % of mental difficulties by sociodemographic and parental characteristics

| | % (N) | SDQ score≥18 % | OR | 95% CI |
|---|---|---|---|---|
| **Sex** | | | | |
| Male | 51.6 (550) | 12.1 | 1 | Ref |
| Female | 48.4 (560) | 15.0 | 1.28 | (0.88 to 1.86) |
| **Age (years)** | | | | |
| 11 | 1.1 (14) | 35.3* | 3.79* | (1.11 to 12.93) |
| 12 | 9.7 (111) | 18.4 | 1.57 | (0.86 to 2.86) |
| 13 | 38.9 (432) | 12.9 | 1.03 | (0.68 to 1.55) |
| 14 | 50.4 (553) | 12.6 | 1 | Ref |
| **Ethnic group** | | | | |
| White British | 86.1 (839) | 14.1 | 1 | Ref |
| Other ethnic group | 13.9 (271) | 9.6 | 0.65 | (0.37 to 1.11) |
| **Parental highest social class (NS-SEC)** | | | | |
| Management and professional | 41.8 (439) | 9.0 | 1 | Ref |
| Intermediate | 22.7 (253) | 14.0 | 1.64 | (0.99 to 2.74) |
| Routine and manual | 31.0 (345) | 17.3** | 2.11** | (1.34 to 3.33) |
| Unemployed | 4.4 (53) | 26.9** | 3.71** | (1.56 to 8.84) |
| **Mother's highest qualification** | | | | |
| Degree or higher | 33.2 (351) | 11.1 | 1 | Ref |
| A-level or equivalent | 17.5 (185) | 11.1 | 1.00 | (0.55 to 1.84) |
| GCSE or equivalent | 29.5 (309) | 13.3 | 1.23 | (0.75 to 2.01) |
| None/other | 19.8 (239) | 20.0** | 2.00** | (1.20 to 3.33) |
| **Household poverty score** | | | | |
| Not at all deprived | 20.9 (179) | 8.0 | 1 | Ref |
| Somewhat deprived | 54.0 (493) | 11.6 | 1.50 | (0.78 to 2.88) |
| Highly deprived | 25.1 (266) | 22.1*** | 3.26*** | (1.67 to 6.36) |
| **Family composition** | | | | |
| Two-parent | 69.7 (759) | 12.0 | 1 | Ref |
| Single parent | 27.8 (321) | 18.5* | 1.66* | (1.12 to 2.47) |
| Other | 2.5 (30) | Suppressed | – | – |
| **Happy with school work** | | | | |
| Happy | 74.7 (840) | 9.0 | 1 | Ref |
| Not happy | 25.3 (263) | 26.8*** | 3.71*** | (2.52 to 5.47) |
| **Happy with school** | | | | |
| Happy | 78.6 (876) | 9.3 | 1 | Ref |
| Not happy | 21.4 (220) | 28.9*** | 3.96*** | (2.66 to 5.90) |
| **Parental interest in school** | | | | |
| Always or nearly always | 79.0 (871) | 10.6 | 1 | Ref |
| Sometimes or rarely | 21.0 (220) | 24.4*** | 2.73*** | (1.8 to 4.10) |
| **Regularly attends parents' evenings** | | | | |
| Always or nearly always | 81.1 (896) | 10.8 | 1 | Ref |
| Sometimes or rarely | 18.9 (199) | 24.9*** | 2.73*** | (1.79 to 4.16) |
| **Feels supported by family** | | | | |
| Always or mostly | 76.3 (837) | 9.0 | 1 | Ref |
| Not supported | 23.7 (269) | 27.8*** | 3.87*** | (2.62 to 5.71) |
| **Regularly quarrels with either parent** | | | | |

**Table 2** Continued

| | % (N) | SDQ score≥18 % | OR | 95% CI |
|---|---|---|---|---|
| Less than once a week | 60.0 (662) | 7.5 | 1 | Ref |
| More than once a week | 40.0 (423) | 22.5*** | 3.59*** | (2.40 to 5.36) |
| Parental mental health | | | | |
| Not poor | 56.8 (539) | 11.3 | 1 | Ref |
| Poor | 43.2 (423) | 16.4* | 1.55* | (1.02 to 2.36) |
| Parental physical health | | | | |
| Not poor | 58.6 (564) | 11.3 | 1 | Ref |
| Poor | 41.4 (402) | 16.6* | 1.57* | (1.04 to 2.37) |
| Attainment at key stage 2 Maths | | | | |
| Achieved level 4 | 71.5 (860) | 12.5 | 1 | Ref |
| Did not achieve level 4 | 17.4 (169) | 18.2 | 1.56 | (0.98 to 2.48) |
| Attainment at key stage 2 Writing | | | | |
| Achieved level 4 | 82.6 (270) | 11.5 | 1 | Ref |
| Did not achieve level 4 | 28.4 (759) | 18.4** | 1.72** | (1.15 to 2.58) |
| Attainment at key stage 2 Reading | | | | |
| Achieved level 4 | 92.3 (947) | 13.4 | 1 | Ref |
| Did not achieve level 4 | 7.7 (74) | 15.1 | 1.15 | (0.56 to 2.37) |

Ref=reference group; unweighted N; imputed and weighted percentages shown; some values are suppressed due to small base sizes and risk of disclosure; ***p<0.001, **p<0.01, *p<0.05.
NS-SEC, National Statistics Socioeconomic Classification; SDQ, Strengths and Difficulties questionnaire.

well-established and significantly lower level of attainment in males than females observed in table 1. There was an independent relationship between poor mental health and low attainment in males after controlling for all explanatory variables (OR 2.77, (1.30 to 6.29)). For females, the relationship between poor mental health and low attainment was no longer significant once prior attainment, sociodemographic factors and school enjoyment and parental support and engagement with school was controlled for.

For both sexes, there were significant and generally strong associations between subdomains of mental health and attainment. The single noteworthy exception was a lack of association with attainment in females with emotional disorder (OR 1.49, (0.91 to 2.43)). With exception to hyperactivity disorder, there were no significant associations with attainment in males and females after adjusting for sociodemographic factors and happiness with school. Hyperactivity disorder predicted poor academic attainment for males (OR 2.17, 95% CI 1.13 to 4.19) and females (OR 2.85, 95% CI 1.24 to 6.03) after controlling for the effects of all explanatory variables.

*Fully adjusted*: odds of low attainment controlling for age, ethnicity, prior attainment at KS2, household social class, maternal education, household poverty, family composition, happy with school work, happy with school, parental interest in school, parents attend parent evening, family support, quarrels with parents, parental mental and physical health. ***p<0.001, **p<0.01, *p<0.05.

## DISCUSSION

This longitudinal sample of adolescents observed a strong association between mental health difficulties between the ages of 11 and 14 and later educational attainment at age 16. After accounting for the confounding effects of a range of socioeconomic, school-based and parenting factors known to predict lower attainment, young people with mental difficulties were two times as likely to not reach the educational benchmark in England.

The association between lower attainment and overall mental difficulties was largely driven by the presence of hyperactivity disorder, which remained highly significant after accounting for other explanatory factors. The relationship between hyperactivity disorder and lower attainment has been documented elsewhere.[35] Our data support the on-going development early interventions targeted towards hyperactivity disorders[36] focussing on meeting the specific needs of children and young people to enable them to reach their academic potential. Importantly, these interventions are and ought to continue to be school based as it offers a suitable medium for universal support and equal access to provision to nearly all young people.[37]

While males and females with overall mental difficulties were equally likely to not achieve the GCSE benchmark, this relationship was only significant for males after controlling for explanatory factors. This is concurrent with previous work on the same sample assessing educational attainment at older ages,[38] which demonstrated

**Table 3** ORs for low attainment at key stage 4 by total mental health difficulties and domain scores, adjusted stepwise for explanatory factors

| | Model 1 | Model 2 | Model 3 | Model 4 | Model 5 | Model 6 | Model 7 |
|---|---|---|---|---|---|---|---|
| Emotional | 1.64 (1.11,2.41)* | 1.88 (1.27,2.78)** | 1.75 (1.07,2.85)* | 1.55 (0.91,2.65) | 1.22 (0.71,2.10) | 1.12 (0.63,1.99) | 1.07 (0.61,1.90) |
| Peer | 2.44 (1.66,3.58)*** | 2.45 (1.66,3.61)*** | 1.67 (1.02,2.75)* | 1.50 (0.88,2.55) | 1.31 (0.78,2.20) | 1.26 (0.74,2.16) | 1.20 (0.70,2.08) |
| Conduct | 1.92 (1.33,2.76)*** | 1.83 (1.26,2.65)** | 1.91 (1.22,3.01)** | 1.65 (1.02,2.67)* | 1.25 (0.74,2.11) | 1.10 (0.62,1.94) | 1.07 (0.60,1.90) |
| Hyperactivity | 2.52 (1.80,3.52)*** | 2.46 (1.75,3.45)*** | 2.77 (1.84,4.18)*** | 2.94 (1.89,4.57)*** | 2.39 (1.52,3.78)*** | 2.35 (1.46,3.78)*** | 2.38*** |
| Total score | 3.11 (2.11,4.57) | 3.25 (2.20,4.80) | 3.55 (2.22,5.70) | 3.20 (1.90,5.37) | 2.38 (1.38,4.12) | 2.10 (1.17,3.77) | 2.05 (1.15,3.68) |

Imputed model, N=1100.
Model 1: unadjusted odds of low KS4 attainment.
Model 2: adjusts for Model 1+age, sex, ethnicity.
Model 3: adjusts for Model 2+prior attainment at KS2.
Model 4: adjusts for Model 3+household social class, maternal education, household poverty, family composition.
Model 5: adjusts for Model 4+happy with school work, happy with school.
Model 6: adjusts Model 5+parental interest in school, parents attend parent evening, family support, quarrels with parents.
Model 7: adjusts for Model 6+parental mental and physical health.
*** p<0.001, **p<0.01, *p<0.05.

**Table 4** Unadjusted and adjusted ORs for low attainment at key stage 4, as predicted by mental health difficulties, stratified by sex

| | | Unadjusted | Fully adjusted |
|---|---|---|---|
| Emotional | Male | 3.07 (1.48 to 6.38)** | 2.36 (0.83 to 6.64) |
| | Female | 1.49 (0.91 to 2.43) | 0.73 (0.34 to 1.57) |
| Peer | Male | 2.36 (1.39 to 4.02)** | 1.79 (0.83 to 3.84) |
| | Female | 2.55 (1.45 to 4.48)** | 0.99 (0.41 to 2.40) |
| Conduct | Male | 1.65 (1.03 to2.66)* | 0.93 (0.42 to 2.05) |
| | Female | 2.17 (1.22 to 3.86)** | 1.29 (0.52 to 3.18) |
| Hyperactivity | Male | 2.35 (1.49 to 3.71)*** | 2.17 (1.11 to 4.23)* |
| | Female | 2.63 (1.59 to 4.35)*** | 2.85 (1.30 to 6.23)** |
| Total score | Male | 3.16 (1.79 to 5.60)*** | 2.77 (1.24 to 6.16)* |
| | Female | 3.36 (1.97 to 5.71)*** | 1.69 (0.72 to 3.95) |

Imputed model, men N=550; women N=560. Unadjusted: unadjusted odds of low attainment. Fully adjusted: odds of low attainment controlling for age, ethnicity, prior attainment at KS2, household social class, maternal education, household poverty, family composition, happy with school work, happy with school, parental interest in school, parents attend parent evening, family support, quarrels with parents, parental mental and physical health. Results for the stepwise adjustment towards the full model are found in online supplemental table A. *** p<0.001,**p<0.01, *p<0.05.

that females at age 18 exhibited a weaker relationship between mental difficulties and attainment than males. However, in contrast to our findings at ages 11–14 years, females at age 18 were significantly more likely to experience poor mental health than males—females being more likely to be conscientious high achievers was suggested as a possible explanation. Although the reason for this difference needs further investigation, these findings confirm important age and sex differences, which ought to be accounted for when devising interventions aimed at promoting adolescent mental health.

It is noteworthy that although family socioeconomic circumstances are well-established predictors of later performance at school,[39] the association with mental health difficulties was robust to adjustment. Although the association between poorer mental health and lower attainment operated regardless of socioeconomic background, interventions to improve mental health delivered via universal and inclusive mainstream or alternative education-based settings are likely to disproportionately impact those from disadvantaged backgrounds as they are more likely to experience mental health difficulties. Based on findings presented here, improving mental health could possibly increase average attainment levels within this group to a greater extent than within the majority population who are not disadvantaged. The potential effect at a population level would be to reduce the average difference in attainment between socioeconomic groups, and narrow educational and consequent social inequalities.

Overall, these data are of interest to a range for stakeholders as they offer a contemporary and contextually rich data useful for wider policymaking and practice. Furthermore, showing the strong association between

social factors with attainment and mental health makes the fully adjusted independent link between mental health and attainment, all the more striking highlighting that they are both important predictors of attainment.

## Limitations

Consent to data linkage and successful linkage between the UKHLS and the NPD was predicted by ethnicity, household structure and social class. The inclusion of these variables in the imputation and the final models may mitigate against some of these selection effects, the lack of an analytic weight and the ethical limitation of being unable to impute missing data for sensitive information which has been actively protected by the respondent means that data may not be representative; prevalence estimates should be interpreted cautiously and may not be generalisable to the English population. This does not, however, diminish confidence in the associations identified by the prospective approach taken. Although the collection of mental difficulty data from young people is preferable than from their parents, this information was self-reported rather than a clinical diagnosis. Other measures of well-being and mental health ought to be considered in future analysis as associations with different constructs may differ from those presented here. Cut points for the SDQ are contested with researchers in different contexts opting for different thresholds. The SDQ developer adds the caveats to a recently devised set of cut points that these systems 'only provide a rough-and-ready way of screening for disorders'.[40] Finally, mediation analysis has not been conducted in this study though predictors of attainment such as happiness with school may be candidate variables. Caution should be applied to interpreting these candidate mediators as current estimates of the effect of mental difficulties on attainment may be considered overadjusted.

**Contributors**  NRS and LM designed the analysis, which was carried out by LM and MA under guidance from MS and NRS. NRS drafted the manuscript. AH and SS contributed to the study design and all were involved in the drafting of the manuscript.

**Funding**  The project was funded by the Economic and Social ResearchCouncil Secondary Data Analysis Initiative (ES/R005400/1 to NS).

**Competing interests**  None declared.

**Patient and public involvement statement**  It was not appropriate or possible to involve patients or the public in the design, or conduct, or reporting, or dissemination plans of our research.

**Patient consent for publication**  Obtained.

**Ethics approval**  The data used are publicly available via UK Data Service repository (study numbers 6614 and 8644), and do not require ethical assessment for academic research purposes. The University of Essex Ethics Committee approved the survey data collection. No ethics approval number was produced. Ethics approval for data collection was granted by letter dated 6 July 2007 for Waves 1 and 2 and by letter dated 17 December 2010 for Waves 3 to 5. https://www.understandingsociety.ac.uk/documentation/mainstage/userguides/main-survey-user-guide/ethics.

**Provenance and peer review**  Not commissioned; externally peer reviewed.

**Data availability statement**  Data are available in a public, open access repository. All data are hosted by the UK Data Service (UKDS): National Pupil Database data is available under secure access licence agreement to registered and approved researchers.10.5255/UKDA-SN-7642-2. Understanding Society Main Survey Data are available to registered users under standard terms of the UKDS End User Licence Agreement. http://doi.org/10.5255/UKDA-SN-6614-13.

**ORCID iDs**
Neil R Smith http://orcid.org/0000-0002-0023-4398
Stephen Stansfeld http://orcid.org/0000-0001-8716-3897

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
