## [Reviewer comments · BMJ Open]

ARTICLE DETAILS

TITLE (PROVISIONAL)	Adolescent mental health difficulties and educational attainment: findings from the UK Household Longitudinal Study
AUTHORS	SMITH, NEIL; Marshall, Lydia; Albakri, Muslihah; Smuk, Melanie; Hagell, Ann; Stansfeld, Stephen

VERSION 1 – REVIEW

REVIEWER	Wickersham, Alice King's College London, Institute of Psychiatry, Psychology and Neuroscience
REVIEW RETURNED	11-Jan-2021

GENERAL COMMENTS	Thank you for the opportunity to review this manuscript, it's a research area I'm particularly interested in so I was delighted to see this work. The findings are interesting and worthwhile. I've suggested some revisions, mainly clarifications, which would strengthen the manuscript. ABSTRACT - If possible within the word limit, it would be worth making brief reference to findings for the SDQ subscales in the abstract. INTRODUCTION - The references are mostly up-to-date and appropriate. One paper I might have expected to see is from Dalsgaard et al. (DOI: 10.1001/jamapsychiatry.2020.0217). - Some of the terminology around mediation, moderation and confounding needs checking and revising, particularly in paragraph 2 of the introduction, where they are muddled in a couple of places. - Evidence for longitudinal associations are not as limited as the authors suggest in paragraph 3 of the introduction, and come from many countries (e.g. see Riglin et al., DOI: 10.1016/j.adolescence.2014.02.010, for review). The novelty of this work lies more in the use of an objective measure of attainment, and the adjustment for prior attainment, so these are the points I'd consider emphasising. - A reference in support of the final sentence of paragraph 3 which I might suggest adding is Clayborne et al. (DOI: 10.1016/j.jaac.2018.07.896) METHODS - A STROBE checklist has been supplemented, but doesn't appear to have been completed or referenced in the main body of the paper. - For those unfamiliar with the UKHLS, what does wave 2 consist of, and why has it not been included for this analysis?
--

	 - It was difficult to track the changing sample size from study population down the final analytical sample - might it be possible to include a flow diagram to support this? - Why was the pro-social behaviour subscale of the SDQ not used? Could the authors cite a precedent for the chosen clinically-relevant cut point? Was the cut point used in analyses? - When does the "age" variable refer to? Is it age at which mental health was measured? - There are multiple variables measuring similar constructs, such as socioeconomic status etc - have checks for multicollinearity been conducted? - The final paragraph of the Explanatory Variables section might belong in the earlier UKHLS section. - Which explanatory variables had missing data? And what variables were used to conduct the MI? - Please provide an ethics statement. - Stratification by gender was conducted but is not mentioned in the Statistical Analysis section. RESULTS  - It would be helpful to open the results section with an overview of characteristics/key descriptives for the sample. - Please introduce the abbreviations "KS2" and "KS4" prior to the results section, or consider using more common language to describe them, such as "prior attainment" and "educational outcome", or similar. - It is not clear to me what purpose the logistic regression analyses in Table 2 (and to some extent Table 1) serve, as they do not seem to address the research question, or to inform the subsequent analyses which do address the research question. Please can this be clarified, and/or perhaps moved to the supplement? - The confidence intervals for the KS2 explanatory variables in Table 1 seem to be very wide, do the authors have an idea of why this might be? DISCUSSION  - Much of the discussion is dedicated to discussion moderators (e.g. gender, socioeconomic status) and mediators (e.g. happiness with school). But it's important to emphasise that the analyses conducted here cannot provide any support for these effects - follow-up analyses like interactions would have been required. The results only show a direct association between mental health and educational attainment after adjusting for important covariates. Implications flowing from moderators and mediators should therefore be minimised, or briefly mentioned as an area for future work. I'd instead focus on the importance of the finding that mental health seems to be robustly linked to lower attainment. - Results for different subscales don't seem to be discussed, it would be interesting to elaborate on these. - There remains the possibility of over-adjustment, as several candidate mediator variables were adjusted for. This should be reiterated in the limitations section to emphasise that the effect sizes could be under-estimated as a result. For an explanation of this, see DOI: 10.1097/EDE.0b013e3181a819a1. - The use of self-reported symptom scales, rather than clinical diagnosis, could be mentioned in the limitations section.
--	--

REVIEWER	Domingue, Ben Stanford University
-----------------	--------------------------------------

GENERAL COMMENTS

The paper in question, "Adolescent mental health difficulties and educational attainment: findings from the UK Household Longitudinal Study", looks at associations between mental health and subsequent educational attainment in a longitudinal design. The authors find strong associations between mental health and subsequent educational attainment. However, there are some key problems with the manuscript in my view. In particular, the results are poorly situated within the broader set of findings on this problem and the analysis itself is challenging to parse. I list these (and other) issues below.

---There is a vast literature on connections between mental health and educational attainment that the authors need to acknowledge. They write "International research has demonstrated numerous associations between mental health and educational attainment (5,13–16)" (p5 l51). That is a start, but it is insufficient. A few minutes of searching yielded many papers that seem relevant to the problem:

<https://doi.org/10.1016/j.psychres.2012.03.040>

<https://doi.org/10.1016/j.jpsychires.2008.01.016>

<https://doi.org/10.1371/journal.pone.0101751>

<https://doi.org/10.1177/0022146512462888>

<https://doi.org/10.1186/s12888-014-0237-4>

My point here is not that the authors need to add these references to their paper. Rather, it is that a more complete accounting of the literature likely makes the question that they are interested in one that we already have a fairly decent answer to. This doesn't mean that the authors have nothing valuable to contribute, but I think that they need to do a better job of modestly characterizing the novelty of their findings in the context of this larger body of research.

---Tables 1 & 2 are effectively presentations of descriptive statistics for a large number of covariates and then associations with either the outcome (Table 1) or the key independent variable (Table 2). I find these tables challenging to interpret; there are too many variables. I wonder if the construction of composites might simplify the narrative. So, for instance, could 'feels supported by family', 'quarrels with parents', 'parental interest in school', and a few others be collapsed into a single composite on parenting?

---Table 3: In these analyses, why dichotomize the outcome and the key predictor? I am sympathetic about the fact that they are both variables that are somewhat messy in that they are counts of related but nonidentical objects. Still, it seems like a great deal of variation is being thrown away by the dichotomization. I wonder whether some analysis could be done so as to construct a more informative set of predictor/outcome variables (e.g., PCA to reduce either predictor or outcome to a single dimension?). It also seems like most of the total score results are driven by hyperactivity, especially in the later models. Is this a finding that merits comment? Finally, it is right that the sample size is 1100 for each cell of that table? What proportion of the outcomes are imputed in each case?

---Table 4: This would be much easier to interpret as a figure. The quantity of results is also overwhelming; I would encourage the authors to concentrate on a single row or column of this table. The full set of results could be presented in a supplement if the authors thought that they had value.

---Initial reaction: Does it make sense to control for happiness with school/schoolwork when you're investigating the link between

	mental health and educational attainment? What are you trying to do when you control for a potential mediator? They address this briefly in the discussion but it seems like an issue that requires more attention. ---I think this sentence means the opposite of what the authors intend. Don't they mean "increases with age"?: The mental health of children and young people in England declines with age with around 14.4% of 11-16 years experiencing a mental disorder compared to 5.5% in their pre-school counterparts aged 2-4 years. p5 l7
--	--

VERSION 1 – AUTHOR RESPONSE

Reviewer: 1

Thank you for your helpful comments which have strengthened the manuscript by excluding some unnecessary content for the reader and identifying where the interpretation could be made clearer.

- If possible within the word limit, it would be worth making brief reference to findings for the SDQ subscales in the abstract.

The following key finding has been added within the strict confines of the word count:

“Hyperactivity disorder strongly predicted lower attainment for males (OR: 2.17 (95% CI: [1.11, 4.23]) and females (OR: 2.85 (95% CI: [1.30, 6.23]).”

INTRODUCTION

- The references are mostly up-to-date and appropriate. One paper I might have expected to see is from Dalsgaard et al. (DOI: 10.1001/jamapsychiatry.2020.0217).

This has been added to the references demonstrating the link between mental ill health and later attainment in a cross-national context.

- Some of the terminology around mediation, moderation and confounding needs checking and revising, particularly in paragraph 2 of the introduction, where they are muddled in a couple of places.

Thank you, on re-reading we agree. This reflects a broader issue covered in later comments (and from other reviewers) that issues around mediation and moderation are not formally tested within this paper and this terminology was incorrectly applied in the original submission. This section has therefore been more accurately framed around the potential for confounding of the relationship between mental health and attainment, with the relationship being explained by a variety of other factors. Mediation and moderation has not been tested, and is therefore not discussed in these terms.

- Evidence for longitudinal associations are not as limited as the authors suggest in paragraph 3 of the introduction, and come from many countries (e.g. see Riglin et al., DOI: 10.1016/j.adolescence.2014.02.010, for review). The novelty of this work lies more in the use of an objective measure of attainment, and the adjustment for prior attainment, so these are the points I'd consider emphasising.

Thank you, this is a very helpful re-framing of the contribution of our work. We have amended our comments on the relatively weak longitudinal evidence base, and emphasised the use of our novel data linkage to objective and nationally representative attainment data as follows:

“Therefore, this study uses the a novel and contemporary data linkage contemporary between the nationally representative UK Household Longitudinal Study linked to objectively measured official education records, to test association poor mental health and poor educational attainment.”

- A reference in support of the final sentence of paragraph 3 which I might suggest adding is Clayborne et al. (DOI: 10.1016/j.jaac.2018.07.896)

A highly appropriate addition, thank you.

METHODS

- A STROBE checklist has been supplemented, but doesn't appear to have been completed or referenced in the main body of the paper.

This has been updated.

- For those unfamiliar with the UKHLS, what does wave 2 consist of, and why has it not been included for this analysis?

Additional text: "Wave 2 (2010-2012) was excluded as it did not ask for information about mental health"

- It was difficult to track the changing sample size from study population down the final analytical sample - might it be possible to include a flow diagram to support this?

Flow chart added

- Why was the pro-social behaviour subscale of the SDQ not used? Could the authors cite a precedent for the chosen clinically-relevant cut point? Was the cut point used in analyses?

The prosocial behaviour scale assesses positive attributes whereas all other domains assess negative or "problem" behaviours. Therefore a lack of prosocial behaviour problems is conceptually different from the presence of psychological difficulties (Goodman, 1997). The prosocial behaviour scale assesses resources rather than problems so it is inappropriate to include it in the total difficulty score. Given it is conceptually different to all other scales and not a part of the total score, it has not been used in this analysis.

Cut points for the SDQ are contested with authors opting for different thresholds. Even the SDQ developer (Goodman) is uncertain of the most appropriate cut points and a new set of cut points has recently been devised. To quote: "both these [scoring] systems only provide a rough-and-ready way of screening for disorders; combining information from SDQ symptom and impact scores from multiple informants is better, but still far from perfect."

For total difficulties scores, we followed the SDQ developer's guidance of coding the top 10% of the population as "abnormal" (has mental health difficulties). The resulting cut point of ≥ 18 also aligns with the recommended cut point based on a standard population, also published by the author Goodman here: <https://www.sdqinfo.org/py/sdqinfo/c0.py>

For individual domain scores, we followed cut points defined by Goodman as "abnormal". These have recently and successfully been used by Deighton et al (2019) in order to provide comparability to a similar English sample. <https://doi.org/10.1192/bjp.2019.19>.

Further detail on these cut points, including the Deighton reference, have been added to the manuscript.

- When does the "age" variable refer to? Is it age at which mental health was measured?

Additional text: "All non-educational attainment measures were taken at the time mental health was assessed."

- There are multiple variables measuring similar constructs, such as socioeconomic status etc - have checks for multicollinearity been conducted?

Yes, in particular the lack of cross-over between poverty score and mother's education and household social class was striking. We expect the overall lack of cross over to be due to different units of analysis being used: household highest, individual level maternal education, and a composite poverty score.

- The final paragraph of the Explanatory Variables section might belong in the earlier UKHLS section.

This has been moved as suggested, thank you.

- Which explanatory variables had missing data? And what variables were used to conduct the MI?

A line has been added to specify that all explanatory variables and outcome variables in table 1 were used in the imputation.

"Complete data was available was available for age, sex, ethnicity and family composition and all variables shown in Table 1 were used in the imputation."

- Please provide an ethics statement.

This has been added to the end of the manuscript.

- Stratification by gender was conducted but is not mentioned in the Statistical Analysis section.

Analysis section has been updated to reflect this.

RESULTS

- It would be helpful to open the results section with an overview of characteristics/key descriptives for the sample.

Table 1 has been updated to convert the unweighted N to weighted %s. Summary text is provided.

- Please introduce the abbreviations "KS2" and "KS4" prior to the results section, or consider using more common language to describe them, such as "prior attainment" and "educational outcome", or similar.

These have been introduced in the explanatory factors (KS2) and Educational Outcomes (KS4) section.

- It is not clear to me what purpose the logistic regression analyses in Table 2 (and to some extent Table 1) serve, as they do not seem to address the research question, or to inform the subsequent analyses which do address the research question. Please can this be clarified, and/or perhaps moved to the supplement?

The logistic regressions in tables 1 and 2 specifically test for significant associations between the risk factors for low attainment or mental difficulties and provide a strength of that association. Without this significance test we would not know that there are no significant ethnic differences in mental difficulties despite there being a sizeable 4.5% percentage difference. We have therefore simplified the table by removing the odds ratios, though we do acknowledge the dependence on p values which are increasingly being considered sub-optimal by journal editors.

With respect to answering the research questions, each table demonstrates that a range of social factors are significantly associated with low attainment and with mental health, justifying their later inclusion in the fully adjusted model. These background data are demonstrably of interest to a range for stakeholders based on preliminary presentation to practitioners and policy-makers, particularly to the Department for Education who have specifically requested their publication in this contemporary and contextually rich data useful for wider policymaking. Furthermore, showing the strong association between social factors with attainment/mental health makes the fully adjusted independent link between mental health and attainment all the more striking highlighting that they are both important predictors of attainment, but that they operate differently.

- The confidence intervals for the KS2 explanatory variables in Table 1 seem to be very wide, do the authors have an idea of why this might be?

The cell count for these groups is relatively small leading to a lower level of precision. e.g. n=74 had low reading scores at KS2, with only 9.5% (91.5-100) of the 74 having high attainment at KS4 giving a small numerator in the calculation of the odds of the exposed "low attainment" group.

DISCUSSION

- Much of the discussion is dedicated to discussion moderators (e.g. gender, socioeconomic status) and mediators (e.g. happiness with school). But it's important to emphasise that the analyses conducted here cannot provide any support for these effects - follow-up analyses like interactions would have been required. ...

We agree, thank you. We have pared back the discussion around mediation and moderation in the introduction and discussion and we have stated that has not been specifically tested for.

- Results for different subscales don't seem to be discussed, it would be interesting to elaborate on these.

A paragraph on the role of the hyperactivity domain in particular has been added, outlining hyperactivity appears to be the main driver of the relationship between attainment and mental difficulties overall.

- There remains the possibility of over-adjustment, as several candidate mediator variables were adjusted for. This should be reiterated in the limitations section to emphasise that the effect sizes could be under-estimated as a result. For an explanation of this, see DOI: 10.1097/EDE.0b013e3181a819a1.

We have addressed this in response to other comments. It was incorrect to suggest that these variables were conceptualised as mediators which is why we consequently did not test for this. School satisfaction/happiness and parental home support/relationships are all considered straightforward confounders with known independent effects on attainment but shown to be associated with mental difficulties in table 1. Though as you have pointed out, and particularly for satisfaction with school life, these may be considered as mediators in future work. The interpretation of happiness/school satisfaction ought be considered cautiously and this has been added to the limitations.

- The use of self-reported symptom scales, rather than clinical diagnosis, could be mentioned in the limitations section.

Self-reported data has been acknowledged as a limitation.

Reviewer 2

---There is a vast literature on connections between mental health and educational attainment that the authors need to acknowledge. They write "International research has demonstrated numerous associations between mental health and educational attainment (5,13–16)" (p5 l51). ... but I think that they need to do a better job of modestly characterizing the novelty of their findings in the context of this larger body of research.

In response, and as the reviewer notes, we have already acknowledged that numerous associations between mental health and educational attainment have been established and demonstrated this knowledge in a paragraph, originally providing 5 references (13% of all references permitted). The paragraph that followed was dedicated to the novelty of our analysis, which has been updated to emphasise that few studies to date apply to the English context, and re-iterates that this analysis was novel in the use of a longitudinal nationally representative probability sample (lacking in much international work) from using a data linkage to educational records in England. In recognition of reviewer input we have added an additional line to explicitly re-iterate the above and emphasise the current policy focus on mental health in schools in the English context which is relatively poorly evidenced in comparison to our international colleagues.

---Tables 1 & 2 are effectively presentations of descriptive statistics for a large number of covariates and then associations with either the outcome (Table 1) or the key independent variable (Table 2). I find these tables challenging to interpret; there are too many variables. I wonder if the construction of composites might simplify the narrative. So, for instance, could 'feels supported by family', 'quarrels with parents', 'parental interest in school', and a few others be collapsed into a single composite on parenting?

We have reduced the table content by removing the odds ratios/ measures of effect.

With regards the composites, we have derived a composite measure of the deprivation score to minimise the number of variables shown as the relevant indicators (material deprivation, living standards and income) capture different elements of the same concept (poverty).

With regards the parent

variables, each of these captures distinctly different constructs, each with established links to educational attainment and association with mental health. Family support and quarrelling represents parental closeness and conflict respectively as used in the recognised Child-Parent Relationship scale (Pianta, 1992) : <https://education.virginia.edu/faculty-research/centers-labs-projects/castl/measures-developed-robert-c-pianta-phd>. Interest in school represents passive engagement compared to more active attendance at parent evenings. To aggregate the four to a composite measure would lose important information on specific parenting behaviours – the relative importance of each behaviour could be used to inform the development of policy interventions to improve child performance.

---Table 3: In these analyses, why dichotomize the outcome and the key predictor? I am sympathetic about the fact that they are both variables that are somewhat messy in that they are counts of related but nonidentical objects. Still, it seems like a great deal of variation is being thrown away by the dichotomization. I wonder whether some analysis could be done so as to construct a more informative set of predictor/outcome variables (e.g., PCA to reduce either predictor or outcome to a single dimension?).

The Strengths and Difficulties Questionnaire has been widely used in UK cohort studies for the past 20 years though unfortunately we are unaware of anyone using it/scoring it in ways others than those recommended by the authors (e.g. by using an alternative reductive approach across various domains). We have chosen to dichotomise as this is the most widely used approach in the UK literature, enabling researchers to more easily compare across studies (e.g. Deighton et al, 2020 as cited in the paper). Furthermore, the binary approach is more policy-relevant and easily

understandable for teachers, practitioners and policy makers who will use this research to help improve mental health and attainment. Audiences understand and are more likely to use a “proportion of young people with poor mental health” or the “likelihood of not passing school if in poor health” in practice rather than a relatively abstract coefficient from the continuous scale. For the purpose of demonstrating the straightforward association between poor mental health and attainment in this paper the binary measure appears to meet the aim. We appreciate more nuanced investigations, and particularly mediation analysis not performed here, will require different approaches which will require continuous measures to maximise the variation in response.

It also seems like most of the total score results are driven by hyperactivity, especially in the later models. Is this a finding that merits comment?

Thank you, we have added a line to highlight this. “The fully adjusted model could not account for the higher odds of not reaching the educational benchmark for those with hyperactivity disorder (OR 2.38, 95% CI [1.48-3.82]), implying that hyperactivity is the behaviour which largely drives the association between mental difficulties scores and lower attainment.”

Finally, it is right that the sample size is 1100 for each cell of that table? What proportion of the outcomes are imputed in each case?

1,100 is the final imputed sample size used in the analysis. This means we had 1,100 cases with information on age, gender, ethnicity but more in-depth information from the questionnaire has random degrees of missing information. Fewer than 10 cases were imputed for each outcome. The variable with the largest proportion of missing was parental mental and physical health (13%). We have considered adding the proportion missing for each variable to table 1, but we have already been advised that the table is difficult to interpret hence we have re-configured it in this draft. The proportion of missing can be derived by the reader by subtracting the N from 1,100. Instead we have expanded the section in the statistical analysis section to describe the low proportion of missing overall.

“Missing values for explanatory variables ranged from 1% to 13% and 0.2% for mental difficulties was imputed.”

---Table 4: This would be much easier to interpret as a figure. The quantity of results is also overwhelming; I would encourage the authors to concentrate on a single row or column of this table. The full set of results could be presented in a supplement if the authors thought that they had value.

We agree the original table is too dense. This table has been reduced to 20% of the original size, showing only unadjusted and adjusted odds ratios. The two models/columns show a very clear distinction between unadjusted and adjusted for males and females. The intermediary models have been added as a supplementary table as advised.

---Initial reaction: Does it make sense to control for happiness with school/schoolwork when you're investigating the link between mental health and educational attainment?

Thank you - We have updated the introduction and the discussion to correct our original framing of the research question. As you and others correctly point out controlling for a mediator will result in over-adjustment; we have not tested for mediation and it was incorrect to suggest this originally. “Happiness with school” is a question appropriately phrased for young people to understand to measure satisfaction with school life. It is not in this instance considered a mediator between mental health and attainment but it is a straightforward confounder – it is associated with mental health but it likely has an independent association with attainment. Literature cited (Guzman et al, 2019) demonstrates that happiness with the school environment is distinct from mental health. We have therefore controlled for happiness as it offers an alternative explanation, being associated with mental health (see table 1), but we do not conceptually consider this scale of satisfaction with school

to directly mediate its impact on attainment. In line with other reviewer comments, we have correctly framed happiness, and a range of other covariates as confounders rather than mediators as the effect of these pathways has not been explicitly investigated and consequently we cannot make claims of mediation.

--I think this sentence means the opposite of what the authors intend. Don't they mean "increases with age"? The mental health of children and young people in England declines with age with around 14.4% of 11-16 years experiencing a mental disorder compared to 5.5% in their pre-school counterparts aged 2-4 years. p5 l7

We appreciate it could be clearer- have amended.

“Mental ill-health in children and young people in England increases with age..”

VERSION 2 – REVIEW

REVIEWER	Wickersham, Alice King's College London, Institute of Psychiatry, Psychology and Neuroscience
REVIEW RETURNED	26-May-2021

GENERAL COMMENTS	Thank you to the authors for this revised manuscript. It is much improved, and most of my comments from the previous round have been addressed. This remains an interesting study, and my remaining suggestions for the methods and results are mainly clarifications. However, I think the discussion still needs some work, as at present some of the implications don't appear to be fully substantiated. METHODS  - Educational attainment: The first sentence under this subheading seems to define attainment the wrong way round, presumably low attainment is defined as NOT achieving 5 A*-C GCSEs. - Explanatory variables: Please could the authors add what the three-tier categories were for occupational social class and mother's highest qualification, just to help clarify what the additional categories are adding. - Statistical analysis: Can the authors add a line or two on what led them to a MAR assumption? - Statistical analysis: Please can the authors clarify phrasing around which variables were imputed, and which variables were used to conduct that imputation? I.e., were the explanatory variables with missing data also imputed, or were they just used to impute missing mental health data? RESULTS  - Table 1: I can't find an explanation of what statistical tests the p values are referring to here. Based on the author's responses to the previous round, I assume they are unadjusted logistic regressions with either GCSE or SDQ as the outcome? This also needs to be explained in the method's statistical analysis section, as at present the only regressions referenced there are the ones in Table 2 with GCSE as the outcome. The authors have justified the purpose of the analyses in Table 1 very nicely in their response to my earlier comments, and it would help if this justification also appeared in the manuscript itself. The authors also acknowledge in their response that reporting p values only is sub-optimal; I
--

	agree, and if these analyses are being kept, I'd strongly suggest re-adding the ORs and 95% CIs.  - Table 1: The categories for KS2 are 'low' and 'not low'. 'Low' is a little subjective; to make this consistent with the methods section, it might be better to say 'Level 4 achieved' and 'Level 4 not achieved'. - Table 3: I wonder if this could be replaced with Supplementary Table A? The information between them overlaps, and there's no harm in giving the full table in the main paper, as it's interesting to see at what point in adjustment the associations diminished in strength. DISCUSSION  - "poor mental health is associated with educational performance to the same extent in young people from more advantaged social backgrounds as it is in those from poorer backgrounds" - this sentence could be slightly rephrased or attenuated, as this would only be demonstrated by an effect modification analysis (e.g. interaction terms). It would be more accurate to say that the association was robust to adjustment for socioeconomic background. - "interventions to improve mental health will disproportionately involve those from disadvantaged backgrounds" - this statement needs substantiating with a citation - just because they are more likely to experience mental health difficulties, that does not mean that their access to treatment will be greater. If anything, the opposite may be true. Without much more support, I'm not convinced by the argument that mental health being a driver of attainment is an avenue to close the social inequality gap. - "However, in contrast to our findings, females at age 18 exhibited a weaker relationship between mental difficulties and attainment than males even though they were significantly more likely to experience poor mental health" - these seem to be the findings here as well, so I'm not sure what the contrast is. - "these findings confirm important age and sex differences" - what are the age differences which are being referred to? I'm assuming those in Table 1, but I'd be very cautious about interpreting this gradient, as it runs directly opposite to the increasing rates of mental health problems typically seen with age which the authors cite in their introduction, and is likely an artifact of only having n=14 young people aged 11. - Overall, the thread of the discussion is a little hard to follow, and I recommend the authors reshape this section to prioritize (1) summarising the findings, (2) comparing the findings to previous work, and then finally (3) the direct implications of the findings, which are primarily the need for targeted educational support and possibly mental health support, especially among boys/those with hyperactivity, to improve educational outcomes. - In their response to my previous round of comments, the authors very helpfully provided an outline of how SDQ cut-points are contested. This seems to be an important point, and could be included in the limitations section.
--	---

VERSION 2 – AUTHOR RESPONSE

Thank you for your continued comments – they have offered unusual level of welcome detail which has added considerable clarity to this manuscript.

METHODS

- Educational attainment: The first sentence under this subheading seems to define attainment the wrong way round, presumably low attainment is defined as NOT achieving 5 A*-C GCSEs.

Thank you, this has been corrected.

- Explanatory variables: Please could the authors add what the three-tier categories were for occupational social class and mother's highest qualification, just to help clarify what the additional categories are adding.

Clarifying text which outlines the specific categories has been added, along with a reference (Rose and Pevalin, 2011) to justify the collapsed categories for NS-SEC.

- Statistical analysis: Can the authors add a line or two on what led them to a MAR assumption?

- Statistical analysis: Please can the authors clarify phrasing around which variables were imputed, and which variables were used to conduct that imputation? I.e., were the explanatory variables with missing data also imputed, or were they just used to impute missing mental health data?

This paragraph has been re-written to clarify that missing data was greatest for the measure of household poverty. MAR was assumed due to the systematically higher levels of missingness among households with poorer explanatory outcomes, but especially so for occupational social class, maternal education, family composition and prior attainment. The new paragraph states that all variables in Table 1 were used in the imputation with complete data for age, sex, ethnicity and family composition being used to impute all missing data for explanatory and outcome (mental difficulties) variables.

RESULTS

- Table 1: I can't find an explanation of what statistical tests the p values are referring to here. Based on the author's responses to the previous round, I assume they are unadjusted logistic regressions with either GCSE or SDQ as the outcome? This also needs to be explained in the method's statistical analysis section...

Logistic regression has been used to significance test. This has now been made clear in the final "statistical analysis" section.

-The authors also acknowledge in their response that reporting p values only is sub-optimal; I agree, and if these analyses are being kept, I'd strongly suggest re-adding the ORs and 95% CIs.

We have reinstated the odds ratios and confidence intervals for the reader on request, thank you. The authors have justified the purpose of the analyses in Table 1 very nicely in their response to my earlier comments, and it would help if this justification also appeared in the manuscript itself. We have added a comment on the practical use of the background/contextual to a range of stakeholders to the closing section of the discussion.

- Table 1: The categories for KS2 are 'low' and 'not low'. 'Low' is a little subjective; to make this consistent with the methods section, it might be better to say 'Level 4 achieved' and 'Level 4 not achieved'.

Thank you – we agree. "not achieving level 4" specified is more informative and less subjective and not beyond the boundaries of understanding for a layperson. The table has been updated to use this new label.

- Table 3: I wonder if this could be replaced with Supplementary Table A? The information between them overlaps, and there's no harm in giving the full table in the main paper, as it's interesting to see at what point in adjustment the associations diminished in strength.

We removed the step-wise coefficients shown in Table 3 (now 4) at the request of another reviewer as it was deemed too much information. Given we have reinstated the odds ratios in Tables 1 and 2 we are conscious that the reader may be overwhelmed with data if we were to reinstate table 3 (now 4) in full. Given the key finding of this gender stratified table is whether each domain remains significant or not, we feel that the concise table of unadjusted v adjusted is sufficient given reporting/space constraints. The detailed information is available to the reader in Supplementary table A. Of course, we can change to the full version if you feel very strongly about this.

DISCUSSION

- "poor mental health is associated with educational performance to the same extent in young people from more advantaged social backgrounds as it is in those from poorer backgrounds" - this sentence could be slightly rephrased or attenuated, ... It would be more accurate to say that the association was robust to adjustment for socioeconomic background.

We have made the suggested change, thank you.

- "interventions to improve mental health will disproportionately involve those from disadvantaged backgrounds" - this statement needs substantiating with a citation - just because they are more likely to experience mental health difficulties, that does not mean that their access to treatment will be greater. If anything, the opposite may be true. Without much more support, I'm not convinced by the argument that mental health being a driver of attainment is an avenue to close the social inequality gap.

Hypothetically at least, universally accessible interventions to improve mental health would have a greater impact on the disadvantaged group as they are more likely to have poor mental health. We have tightened up the wording in this section to reflect this ideal scenario and the latter paragraph suggests continued use of educational settings (in mainstream or alternative provision) as well suited to offer universal and inclusive access as a mechanism to support this conjecture. The previous paragraph advocated delivery of interventions via schools based on success documented by the literature, and the introduction refers to school-based delivery as well. The argument is a logical one based on better mental health=better attainment, so that the group with the poorest mental health are likely to see the biggest gains in attainment at a group level.

- "However, in contrast to our findings, females at age 18 exhibited a weaker relationship between mental difficulties and attainment than males even though they were significantly more likely to experience poor mental health" - these seem to be the findings here as well, so I'm not sure what the contrast is.

See below, which is the same issue. (The contrast refers to the higher prevalence of mental difficulties in females at older ages)

- "these findings confirm important age and sex differences" - what are the age differences which are being referred to? I'm assuming those in Table 1, but I'd be very cautious about interpreting this gradient, as it runs directly opposite to the increasing rates of mental health problems typically seen with age which the authors cite in their introduction, and is likely an artifact of only having n=14 young people aged 11.

Apologies, the original text was not clear. We have amended to make clear we are comparing the lack

of gender difference in our sample at 11-14 years compared to a significantly higher prevalence of mental difficulties in females than males at age 18.

- Overall, the thread of the discussion is a little hard to follow, and I recommend the authors reshape this section to prioritize (1) summarising the findings, (2) comparing the findings to previous work, and then finally (3) the direct implications of the findings, which are primarily the need for targeted educational support and possibly mental health support, especially among boys/those with hyperactivity, to improve educational outcomes.

The discussion has been re-ordered in response. It opens with (1) a summary of the main finding; compares these against existing literature (2) to highlight possible drivers of this finding and how these might be managed by interventions, potentially via schools (3). We additionally close with recognition that these contemporary data provide useful public health intelligence to policy-makers and practitioners (included by request).

- In their response to my previous round of comments, the authors very helpfully provided an outline of how SDQ cut-points are contested. This seems to be an important point, and could be included in the limitations section.

The SDQ author's caveat on the use of the cut points has been added to the limitations section. (It's a shame more manuscripts using the SDQ don't acknowledge differing cut points in the literature and research-base).

VERSION 3 – REVIEW

REVIEWER	Wickersham, Alice King's College London, Institute of Psychiatry, Psychology and Neuroscience
REVIEW RETURNED	21-Jun-2021

GENERAL COMMENTS	Many thanks to the authors for their revisions, they have done a good job of addressing my various concerns. As a final point, please could the authors double check the title/footnotes for Table 4. In the table title, it could be clearer which variable(s) are exposures, outcomes, and stratifiers (i.e. unadjusted and adjusted odds ratios for low attainment, as predicted by mental health difficulties, stratified by sex). In the footnotes, 'odds of mental health difficulties' is not quite accurate; the figures are showing odds ratios for low attainment. In the discussion, I still wonder whether the potential for mental health interventions to narrow social inequalities might be overstated, but I can see the point that the authors are trying to make, and am happy to leave this to the editor's discretion.
--

VERSION 3 – AUTHOR RESPONSE

Thank you for your on-going engagement with our findings. We have made the following edits as requested.

Please could the authors double check the title/footnotes for Table 4. In the table title, it could be clearer which variable(s) are exposures, outcomes, and stratifiers (i.e. unadjusted and adjusted odds ratios for low attainment, as predicted by mental health difficulties, stratified by sex).

This has been edited as requested.

In the footnotes, 'odds of mental health difficulties' is not quite accurate; the figures are showing odds ratios for low attainment.

Apologies and thank you, this has been edited.